# A Sexually Dimorphic Role for STAT3 in Sonic Hedgehog Medulloblastoma

**DOI:** 10.3390/cancers11111702

**Published:** 2019-11-01

**Authors:** Christine L. White, W. Samantha N. Jayasekara, Daniel Picard, Jasmine Chen, D. Neil Watkins, Jason E. Cain, Marc Remke, Daniel J. Gough

**Affiliations:** 1Centre for Cancer Research, Hudson Institute of Medical Research, 27–31 Wright St, Clayton, Victoria 3168, Australia; christine.white@hudson.org.au (C.L.W.); samantha.jayasekara@hudson.org.au (W.S.N.J.); nwatkins@cancercare.mb.ca (D.N.W.); jche216@student.monash.edu (J.C.); Jason.cain@hudson.org.au (J.E.C.); 2Department of Molecular and Translational Sciences, Monash University, Clayton, Victoria 3800, Australia; 3Department of Pediatric Neuro-Oncogenomics, German Cancer Research Center (DKFZ), 69120 Heidelberg, Germany; Daniel.Picard@med.uni-duesseldorf.de (D.P.); Marc.Remke@med.uni-duesseldorf.de (M.R.); 4German Cancer Consortium (DKTK), Partner Site Essen, 45147 Düsseldorf, Germany; 5Department of Pediatric Oncology, Hematology, Clinical Immunology, Institute of Neuropathology, Medical Faculty, University Hospital Düsseldorf, 40225 Düsseldorf, Germany; 6Research Institute in Oncology and Hematology, Cancer Care Manitoba, Winnipeg, MB R3E 0V9, Canada; 7Department of Internal Medicine, Rady Faculty of Health Sciences, University of Manitoba, Winnipeg, MB R3T 2N2, Canada

**Keywords:** STAT3, sexual dimorphism, medulloblastoma, sonic hedgehog

## Abstract

Medulloblastoma is the most common malignant brain tumor in children and represents 20% of all pediatric central nervous system neoplasms. While advances in surgery, radiation and chemotherapy have improved overall survival, the lifelong sequelae of these treatments represent a major health care burden and have led to ongoing efforts to find effective targeted treatments. There is a well-recognized male bias in medulloblastoma diagnosis, although the mechanism remains unknown. Herein, we identify a sex-specific role for the transcription factor Signal Transducer and Activator of Transcription 3 (STAT3) in the Sonic Hedgehog (SHH) medulloblastoma subgroup. Specific deletion of *Stat3* from granule cell precursors in a spontaneous mouse model of SHH medulloblastoma completely protects male, but not female mice from tumor initiation. Segregation of SHH medulloblastoma patients into high and low STAT3 expressing cohorts shows that low STAT3 expression correlates with improved overall survival in male patients. We observe sex specific changes in *IL-10* and *IL-6* expression and show that IL-6 stimulation enhances SHH-mediated gene transcription in a STAT3-dependent manner. Together these data identify STAT3 as a key molecule underpinning the sexual dimorphism in medulloblastoma.

## 1. Introduction

Medulloblastoma (MB) is the most common malignant pediatric brain tumor [1]. Improvements in surgery, radiotherapy and chemotherapy have dramatically improved patient outcomes, but are associated with devastating impacts on normal development, with >90% of survivors requiring long-term special education services [2]. Characterization of the genomic, transcriptomic, epigenetic and proteomic landscape of MB have revealed that MB is not a single monolithic disease and is now segregated into at least four disease subgroups: Wingless (WNT); Sonic Hedgehog (SHH); Group 3 and Group 4 [3,4,5]. Subgroups are distinguished by cell of origin, genetic drivers, demographics and prognosis [1,6]. This refined understanding of the disease is now influencing patient care. Indeed, the subgrouping of MB patients now forms the basis of the World Health Organization diagnostic criteria [7] and has led to a surge in subgroup specific, and precision medicine clinical trials targeting the genome-defined vulnerabilities of each subgroup [8]. Male sex is a risk factor for MB with a distribution of males to females around 1.5 to 1 depending on subgroup [9]. This sex bias is most prominent in group 4, where male cases are around 3 times more common than female cases. Not only is there a male bias in the development of MB but male patients also have worse overall survival than females [10]. To date, no study has identified the mechanism that underpins this bias.

MB is a developmental cancer caused by a defect in the programmed proliferation, migration and apoptosis that occurs during normal cerebellar development. This is reflected in the spectrum of genetic mutations that define each MB subgroup. WNT, SHH and MYC signaling are required for embryonic brain development. Signal Transducer and Activator of Transcription (STAT) 3 is a potent transcription factor that initiates gene transcription downstream of Janus Kinases (JAK) in response to cytokine or growth factor stimulation [11,12]. Within the central nervous system (CNS) STAT3 supports proliferation and prevents terminal differentiation of neural precursor cells and radial glia [13,14,15,16]. Diminished STAT3 expression results in terminal differentiation of olfactory bulb neurons in vitro, while high STAT3 expression maintains stem-like properties of neuronal cells in animal models [17]. In contrast, the critical astrocytic differentiation gene, astroglial differentiation and glial fibrillary acidic protein (GFAP) is a STAT3 target gene [18,19]. 

Aberrant STAT3 activity is observed in around 50% of all cancers including those of the CNS. In adult glioblastoma multiforme, elevated STAT3 activity is attributed to enhanced Vascular Endothelial Growth Factor (VEGF) signaling and angiogenesis [20] and correlates with worse prognosis [21]. In MB, STAT3 activation was reported in the CD133^+^ stem cell population in group 3 MB cell lines and inhibition of STAT3 increased survival in mouse models [22]. To date, there have been no studies of the role of STAT3 in other MB subgroups. However, the observation that STAT3 is required for the development of basal cell carcinoma driven by augmented SHH signaling [23] suggests that STAT3 will be important in the SHH MB subgroup.

In this manuscript, we used a spontaneous mouse model of SHH MB to show that deletion of *Stat3* specifically from the granule cell precursors completely protects male mice from developing MB, but has no impact on female mice. Importantly, analysis of a patient cohort shows that *STAT3* expression correlates with patient outcome in males, but not females. At a molecular level, we report a loss of SHH signaling in the absence of *Stat3*. Together, these data reveal that STAT3 expression is a potential prognostic marker of patient outcome in SHH MB patients.

## 2. Results

### 2.1. STAT3 Expression in SHH MB Contributes to Disease Onset in Male but Not Female Mice 

The granule cell precursors (GCP) are the cell of origin of SHH medulloblastoma which, in the mouse, proliferate and migrate to form the granule cell layer by postnatal day (P) 14 [24]. We performed an immunohistochemical analysis of STAT3 expression in the cerebellum of adult wild-type mice and in tumors formed in the *Ptch1^LacZ/+^* mouse model of SHH MB in which exon 1 and 2 of the *Ptch1* gene are replaced with a LacZ reporter resulting in loss of function [25]. Approximately 25% of *Ptch1^LacZ/+^* mice spontaneously develop MB over the course of approximately six months due to sustained SHH signaling [25]. In normal brain we observed that Stat3 expression was restricted to the Purkinje cells, with no expression detected in the granule cell layer (Figure 1). In contrast, we observed increased Stat3 expression stochastically distributed throughout SHH medulloblastoma tumors that arise from the granule cell layer. Moreover, Stat3 staining in tumor sections was predominantly nuclear, indicating transcriptional activity (Figure 1). STAT3 transcriptional activity is dependent on phosphorylation on Y705 and to a lesser extent S727 [26]. Consistent with the nuclear localization of Stat3 in this pool of tumor cells we observed phosphorylation on both Y705 and S727 which was not observed in the granule cell layer of healthy cerebellum (Figure 1). Given that the elevated Stat3 expression was not uniform throughout the tumor and that Stat3 is a critical immune signaling protein we assessed the extent of tumor infiltrating leukocytes. However, we observed a lack of CD45 positive cells in tumors indicating that Stat3 expression was restricted to tumor cells (Appendix A). 

To determine the functional significance of elevated Stat3 expression in SHH MB, we crossed *Ptch1^LacZ^*^/+^ mice with *Math1^Cre^Stat3^LoxP/LoxP^* mice. Expression of Cre recombinase under the control of the *Math1* promoter drives bi-allelic deletion of *Stat3* from GCPs of the cerebellum from embryonic day 11, which precedes MB development in this model [27]. The efficiency of Cre-mediated deletion of *Stat3* was confirmed by genotyping PCR (Appendix A). Mice were monitored for disease onset by the occurrence of hydrocephaly and motor co-ordination deficits and disease confirmed histologically. Comparison of disease onset and overall survival between *Ptch1^LacZ/+^* with *Math1^Cre^Stat3^LoxP/LoxP^Ptch1^LacZ/+^* mice across the entire cohort revealed no statistically significant difference, although a trend towards improved survival was observed for *Math1^Cre^Stat3^LoxP/LoxP^Ptch1^LacZ/+^* mice at later time points (Figure 2A). However, when mice were segregated on the basis of sex, we observed a very dramatic phenotype. While female *Ptch1^LacZ/+^* and *Math1^Cre^Stat3^LoxP/LoxP^Ptch1^LacZ/+^* mice showed no survival differences (Figure 2B), male *Math1^Cre^Stat3^LoxP/LoxP^Ptch1^LacZ/+^* mice showed substantially improved survival relative to male *Ptch1^LacZ/+^* controls (Figure 2C). Indeed, only 2 out of 22 male *Math1^Cre^Stat3^LoxP/LoxP^Ptch1^LacZ/+^* mice developed MB and genotyping analysis of the tumors that arose in these two mice revealed that they were heterozygous for *Stat3* and therefore underwent incomplete recombination. Together, these data reveal that MB tumors fail to form in male mice when STAT3 is deleted from granule cell neurons.

### 2.2. High STAT3 Expression Correlates with Worse Overall Survival in Male SHH MB Patients

To determine whether the Stat3-dependent protective effect we observed in male SHH MB mice translates into patients, we analyzed the impact of *STAT3* expression on survival in data from a publicly accessible cohort of 83 human SHH MB samples [28]. Patients were stratified on the basis of high (greater than the dataset mean) and low (less than the dataset mean) *STAT3* expression and/or sex. *STAT3* expression did not correlate with any improvement in overall survival across the entire cohort (Figure 3A). Female patients with high levels of STAT3 expression trended towards improved survival, especially at extended time-points; however, this did not reach statistical significance (Figure 3B). In contrast, and in close agreement with our data from mouse models, males with low STAT3 expression had significantly improved overall survival (Figure 3C). 

### 2.3. STAT3 Expression Has No Impact on Proliferation, Cell Death or Differentiation

STAT3 drives the expression of genes that control proliferation, apoptosis and differentiation, although to our knowledge this has not been reported to have a sex bias [29]. Therefore, we performed an immunohistochemical analysis of proliferation (PCNA) and apoptosis-associated DNA fragmentation (TUNEL) in tumor tissue from male and female *Ptch1^LacZ/+^* and *Math1^Cre^Stat3^LoxP/LoxP^Ptch1^LacZ/+^* mice. We observed a very high proliferative index with limited cell death which was unchanged between genotypes or sexes (Appendix A). To determine whether *Stat3* deletion from GCP cells altered differentiation of central nervous system cell lineages in mice we analyzed the expression of lineage specific differentiation markers for glia (Gfap), developing neurons (βIII Tubulin) and terminally differentiated neurons (NeuN). We did observe a subtle increase in Gfap in the tumors from *Math1^Cre^Stat3^LoxP/LoxP^Ptch1^LacZ/+^* mice of both sexes, suggesting increased glial differentiation (Appendix A). However, there was no change in developing (Appendix A) or mature (Appendix A) neurons in the tumors from *Math1^Cre^Stat3^LoxP/LoxP^Ptch1^LacZ/+^* mice of either sex.

### 2.4. STAT3 Is Required for SHH Signaling

To determine the molecular basis underlying the improved survival of low-STAT3 male SHH MB mice and patients, we investigated the impact of Stat3 removal on SHH signaling. The primary cilia are a critical cellular organelle for the SHH signaling pathway [30]. SHH engages with its receptor, Patched1 (PTCH1), which releases its repression of Smoothened (SMO), allowing accumulation of SMO within cilia and activation of GLI transcription factors (Gli1, Gli2 and Gli3) [31]. We analyzed the formation of primary cilia by immunofluorescent detection of the cilia marker Arl13b in tissue sections from tumors and found that cilia formation was not dependent on Stat3 expression (Figure 4A, quantified in Figure 4B). In contrast, we did observe sex- and genotype- dependent differences in the transcriptional outputs of SHH signaling. Quantitative real-time PCR (qRT-PCR) analysis of SHH-induced transcripts (*Gli1*, *Gli2*, *Gli3* and *Ptch1*) in tumor tissue isolated from male or female *Ptch1^LacZ/+^* and *Math1^Cre/+^Stat3^LoxP/LoxP^Ptch1^LacZ/+^* mice revealed significant increases in the expression of *Gli1* and *Gli2* transcripts in female *Math1^Cre/+^Stat3^LoxP/LoxP^Ptch1^LacZ/+^* tumors compared to female *Ptch1^LacZ/+^* controls (Figure 4C), suggesting that Stat3 represses SHH signaling in female tumors. However, this does not result in increased tumor incidence or decreased latency in these mice (Figure 2B). In contrast, *Stat3*-deficient male mice showed a statistically significant decrease in *Gli1* and a trend towards decreased *Gli2* expression (Figure 4C). These data suggest that in this disease-relevant setting of constitutive SHH signaling, STAT3 suppresses SHH signaling in females and augments SHH signaling in males. 

To confirm this observation in the context of acute SHH signaling, we stimulated male *Stat3* wild-type and *Stat3*-deficient mouse embryo fibroblasts (MEFs) with SHH and determined the expression of *Gli1*, *Gli2*, *Gli3* and *Ptch1* transcripts. We observed robust increases in the expression of each transcript in SHH stimulated WT MEFS (Figure 4D). In contrast, SHH-induced expression of these factors was completely lost in *Stat3*-deficient MEFs (Figure 4D). Together, these data suggest that while STAT3 is not required for cilia formation, and therefore the apical component of SHH signaling, it is required for the expression of SHH target genes.

### 2.5. IL-6 Expression Is Elevated in Males and Augments SHH Signaling

Our data show that Stat3 expression and activation is elevated in SHH MB and that Stat3 is required for SHH signaling in SHH MB tissue from male mice. Therefore, to determine whether there is any sex- or Stat3-dependent change in the expression of Stat3 activating cytokines or growth factors, we performed a qRT-PCR array for 84 cytokines and chemokines on primary tumor tissue, which was compared to transcriptomics data from SHH MB patients. We observed increased expression of Interferon (IFN)γ, Interleukin (IL)-10 and IL-6 in both mice and humans with high STAT3 expression (Figure 4E). However, only the expression of *IL-10* and *IL-6* was consistently increased in male mice and patients (Figure 4F). To determine whether STAT3 activation by IL-6 or IL-10 enhances SHH signaling we stimulated MEFs with IL-6 or IL-10 alone or in combination with SHH and observed an additive effect of IL-6, but not IL-10 on SHH signaling (Figure 4G). Together these data show that Stat3 is required for SHH signaling and that IL-6 expression is elevated in males which enhances SHH signaling and may contribute to the Stat3 and sex-dependent onset of medulloblastoma.

## 3. Discussion

A well-recognized but under-studied facet of medulloblastoma is the male bias in its presentation. Immunohistochemical analysis of patient tissue and cell lines identified a higher proliferative and lower apoptotic rate in male MB patient tissue and cell lines [32,33]. However, these studies were not stratified on the basis of the current molecular subgroups which may explain why we observed equivalent proliferation and apoptosis in both sexes in mouse models of SHH-medulloblastoma. Our data provides the first description of a molecular target underpinning the sexual dimorphism in MB. Specifically, we reveal that the loss of *Stat3* expression from the cell of origin in a mouse model of SHH MB completely protects male mice from developing disease while having no impact in female mice. Moreover, analysis of patient outcome data on a cohort of SHH MB patients stratified on the basis of *STAT3* expression revealed the same male-specific protection. Finally, we show both in tumor tissue and in cell lines that Stat3 supports SHH signaling which is potentiated by co-treatment with the STAT3 activating cytokine IL-6.

Medulloblastoma is a cancer of brain development, and there are differences in brain development and in CNS pathologies between males and females driven in part by expression of sex hormones [34,35,36]. Intriguingly, STAT3-dependent activities in brain and heart have been linked to estrogen. Estrogen is protective in a mouse model of brain ischemia due, in part to STAT3 activation [37]. Similarly, activation of STAT3 by testosterone was also reported in animal models of heart ischemia [38]. However, it is unlikely that STAT3 is directly phosphorylated in response to testosterone or estrogen stimulation, rather that STAT3 is activated following the release of inflammatory cytokines after ischemic injury [39]. A sexually dimorphic requirement for STAT3 in cancer was observed previously. Hepatocellular carcinoma is a disease predominantly affecting males, which is due in part to the ability of estrogen to suppress IL-6. This results in elevated IL-6 secretion and persistent STAT3 activation in males [22]. This observation is in close agreement with the data presented in this study, which show elevated *IL-6* and *IL-10* expression in male SHH patients and mice. In addition, the co-stimulation with IL-6 and SHH enhances SHH signaling. However, it remains to be determined whether this is through direct crosstalk between STAT3 and SHH signaling, or whether these pathways operate in parallel. A recent proteomic analysis of brains from male and female mice with neuronal specific STAT3 deletion showed a significant reduction in the expression of metabolic and detoxifying proteins including Voltage Dependent Anion Channel (VDAC)2, Manganese Superoxide Dismutase (MnSOD) and Isocitrate Dehydrogenase (IDH) [40]. Given that STAT3 has recently been shown to play a critical role in mitochondrial and metabolic activity in the development of cancers [41,42], it would be intriguing to determine whether STAT3-dependent metabolic changes contribute to sexual dimorphism in SHH MB. 

Together, the data in this study show that STAT3 is required for SHH-mediated gene transcription, and that stimulation of cells with STAT3 activating cytokines potentiates SHH-signaling. Meanwhile, it remains to be seen whether STAT3 expression dictates patient response in a sex-specific manner in the other MB subgroups. Our data suggest that stratification of male SHH MB patients based on STAT3 expression could be an additional clinical tool to identify patients that would be eligible for less aggressive therapies to improve quality of life while still obtaining current overall survival rates. Moreover, it highlights the critical importance of equal distribution of sexes in both pre-clinical and clinical oncology studies.

## 4. Materials and Methods

### 4.1. Mice

Mice were maintained at the Monash Medical Centre Animal Facility under specific pathogen-free conditions in accordance with Australian Government and Monash University guidelines and experimental protocols approved by the Monash Medical Centre Ethics Committee (MMCA-2013-19 Defining the role of STAT3 in Sonic Hedgehog Medulloblastoma). *Stat3^LoxP/LoxP^* [43], *Ptch1^LacZ/+^* [25] (JAX 03081) and *Math1^Cre^* [27] (JAX 011104) mice were intercrossed to generate *Math1^Cre^Stat3^LoxP/LoxP^Ptch1^LacZ/+^* and control cohorts of mice. Mice showing symptoms consistent with SHH MB were sacrificed and the brain was analyzed histologically to confirm the presence of disease.

### 4.2. Bioinformatic Analysis of Human Tissue Samples

Publicly available dataset GSE50765 [28] was imported using the Partek Genomic Suite (Partek Inc., St Louis, MO, USA) using Robust Multi-array Average normalization. Samples with STAT3 signal above the mean were designated ‘high’ STAT3, those below the mean were designated ‘low’ STAT3.

### 4.3. Cell Culture

Mouse embryo fibroblasts were cultured in high glucose (4.5 g/L) DMEM supplemented with 10% Bovine Calf Serum, Sodium Pyruvate and L-Glutamine at 37 °C and 5% CO_2_. MEFs were serum starved in DMEM supplemented with 0.2%BCS for 24 h prior to stimulation with SHH (100 ng/mL final concentration, R&D Systems, 1845-SH-100; 100 ng/mL IL-6 Peprotech, 200–06, 50 ng/mL IL-10, Peprotech, 210–10) for 24 hours prior to harvesting for qRT-PCR analysis.

### 4.4. Immunohistochemistry

Freshly isolated tissue was drop fixed in 10% Neutral Buffered Formalin for up to 48 h before paraffin embedding, sectioning and mounting 4-μM-thick sections. Immunohistochemical sections were dewaxed and rehydrated through an ethanol gradient before antigen retrieval in 10 mM citrate pH 6.0. The following antibodies and concentrations were used: anti-STAT3 (1/50, polyclonal H-190, Santa Cruz Biotechnology catalog #sc-7179), anti STAT3 pY705 (1/50 Cell Signaling Technology catalog #9145), anti-STAT3 pS727 (1/50 Cell Signaling Technology catalog #9134), anti-GFAP – 1/200, Millipore clone GA5, catalog #MAB360), anti-β-III Tubulin (1/200, Millipore catalog # MAB5564) Anti-NeuN, (1/100, clone A60, Millipore catalog # MAB377), anti-CD45 (1/100, BD Biosciences catalog #550539), anti-PCNA (1/200, DAKO catalog M0879). To identify primary cilia sections, they were co-stained with ARL13B (1/200, ProteinTech, catalog # 17711-1AP) and alpha-tubulin directly conjugated to AlexaFluor 488 (1/200, Invitrogen clone B-5-1-2, ThermoFisher catalog # 322588).

For immunohistochemistry, the Vectastain Elite ABC HRP kit (Vector Laboratories, Burlingame, CA, USA) was used for detection following the manufacturer’s instructions. Slides were counterstained with hematoxylin (Dako, Santa Clara, CA, USA) and Scott’s Blue solution dehydrated with ethanol and mounted in DPX mounting medium (Sigma Aldrich, St. Louis, MO, USA).

For immunofluorescence, tissue sections were blocked with 5% normal goat serum (rabbit primary) or the Mouse on Mouse Immunodetection Kit (Vector Laboratories) was used as per the manufacturer’s instructions (mouse primary) and detected with AlexaFluor 488 or AlexaFluor 594 conjugated secondary antibodies (1/1000, Invitrogen, Carlsbad, CA, USA). Samples were mounted in VectaShield Hardset Mounting Medium (Vector Laboratories) and stored at 4 °C. 

The Apoptag Peroxidase In Situ Apoptosis Detection Kit (EMD Millipore, Temecula, CA, USA) was used for TUNEL staining according to manufacturer’s instructions.

Slides were scanned using an Arperio Scanscope (Leica Biosystems Imaging, Wetzlar, Germany) at 20× magnification or visualized using and Nikon DS-Fil H550S brightfield photomicroscope with Nikon DS-5M-L1 camera (Nikon, Tokyo, Japan,). Immunofluorescent stains were visualized using an Olympus FV1200 Confocal microscope (Olympus, Tokyo, Japan).

### 4.5. PCR Array

RNA expression levels of 84 cytokine and related genes were assessed using the RT^2^ Profiler PCR Array (Mouse cytokines and chemokines, Qiagen, Hilden, Germany). RNA was isolated from 10 mg of tumor tissue using the Bioline Isolate II RNA Mini Kit. Following Agilent Bioanalyser quality confirmation, 400 ng of RNA was reverse transcribed using the RT^2^ First Strand synthesis kit (Qiagen), cDNA was applied to the RT^2^ Profiler Array and amplified using a QuantStudio 6 (Life Technologies, Carlsbad, CA, USA) Real Time PCR machine. Manufacturers’ protocols were followed for all kits described.

### 4.6. RNA Isolation and Gene Expression Analyses

Total RNA was isolated from mouse tumor tissue or MEF cultures using the RNAeasy Isolation Kit (Qiagen). cDNA was synthesized using the Superscript® III First-Strand Synthesis System (Invitrogen^TM^) or Transcriptor High Fidelity cDNA Synthesis Kit (Roche, Mannheim, Germany) using OligodT primers according to manufacturer’s instructions. qRT-PCR was performed using SYBR green on the QuantStudio 6 Real Time PCR System (Life Technologies). Transcript levels relative to *B2m* were calculated using the standard curve method and the following primer sequences: Mouse *B2* F 5’- ATGCTGAAGAACGGGAAAAA, R 5’ AGTCTCAGTGGGGGTGAAT; Mouse *Gli1* F 5’ – TGTGTGAGCAAGAAGGTTGC, R 5’ – ATTGGAGTGGGTCCGATTCT; Mouse *Gli2* F 5’ -CAACGCTACTCTCCCAGAC, R 5’ -GAGCCTTGATGTACTGTACCAC; Mouse *Gli3* F 5’-CGAGAACAGATGTCAGCGAG, R 5’ -TGTTGAGACCCTGCACACTC; and Mouse *Ptch1* F 5’-TTCTCGACTCACTCGTCGAC, R 5’-GCAAGTTTTTGGTTGTGGGT.

### 4.7. Mouse Genotyping PCR

DNA for PCR to confirm *Stat3* deletion was extracted from tumor tissue by digestion at 55 °C for 4 hours in a solution of phosphate buffered saline containing 0.045% (*v*/*v*) Triton-X 100, 0.045% (*v*/*v*) IGEPAL-CA630 and 1 mg/mL proteinase K (Worthington Biochemicals, Lakewood, NJ, USA) followed by centrifugation at 10,000× *g*. Tail tissue was digested in 300 µL 50 mM NaOH at 95 °C before neutralization with 100 µL 0.5 M TRIS-HCL (pH = 8.0). STAT3 primers were: ST3B1: CGTGAAAGGCTGAGAAATGCTG; 30163L: GAAGGCAGGTCTCTCTGGTGCTTC’ 14070L: CAGAACCAGGCGGCTCGTGTGCG. Cycling was 94 °C for 3 min, (92 °C 30 s, 65 °C 30 s, 72 °C 45 s) × 40, 72 °C for 7 min. Expected product sizes from this reaction were as follows: *LoxP*-flanked *Stat3* that had not undergone Cre-mediated deletion—350 bp; Cre-deleted *LoxP-* flanked *Stat3*—550 bp; wild-type *Stat3*—250bp.

### 4.8. Statistics

Kaplan-Meier survival curves were generated and compared using the Mantel-Cox Log Rank test with GraphPad Prism software (Graphpad, San Diego, CA, USA). Student’s *t*-test or two-way ANOVA with Tukey’s multiple comparisons test was used to compare data presented in Figure 4B–D.

## 5. Conclusions

The data presented in this study identify a novel sex-specific role for STAT3 in SHH MB. Analysis of a spontaneous mouse model of SHH MB and human SHH patients revealed a pro-tumorigenic role for STAT3 in males but not females. Further investigation identified no role for STAT3 in cilia formation and the upstream elements of SHH signaling. In contrast, STAT3 did was require for efficient transcription of SHH induced gene expression. Elevated IL-6 and IL-10 transcript levels also correlated with male sex and higher levels of STAT3 in both mouse models and human SHH patients, and IL-6 enhanced SHH-mediated gene transcription. These data provide a bases for further analysis of risk stratification of SHH MB patients based on sex and STAT3 expression.

## Figures and Tables

**Figure 1 cancers-11-01702-f001:**
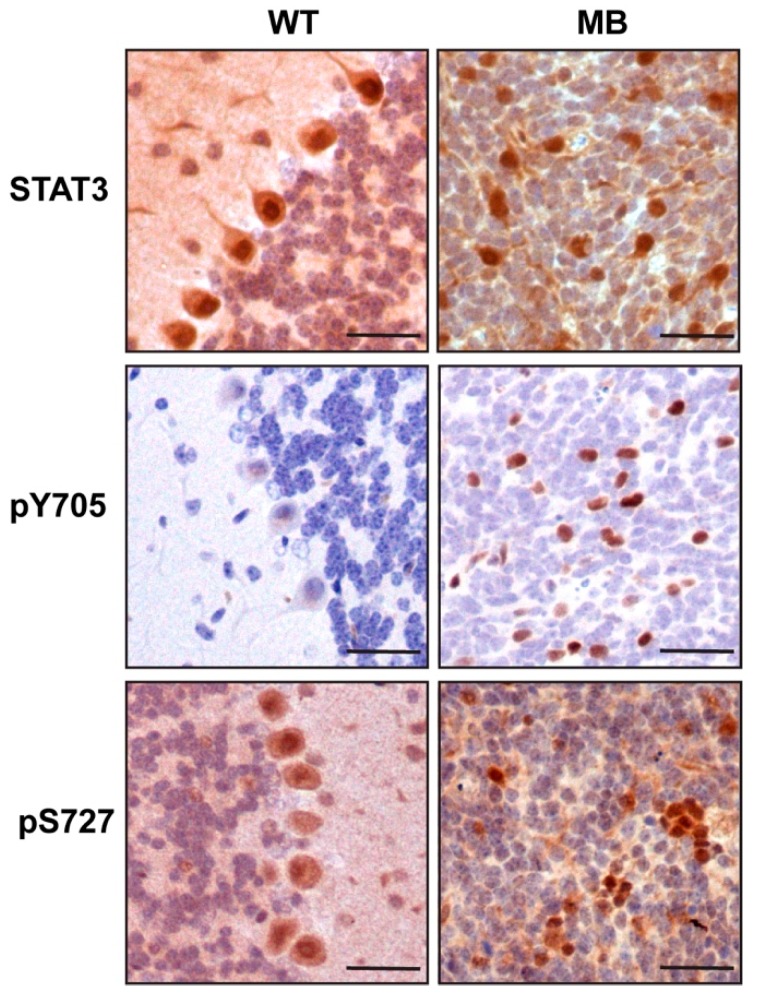
Elevated STAT3 expression and phosphorylation in SHH MB. Immunohistochemical staining for total STAT3, pY705 STAT3 or pS727 STAT3 in normal cerebellum and SHH medulloblastoma tumors from Ptch1^LacZ/^^+^ mice. Representative fields from at least three independent mice. Scale bar = 100 μm.

**Figure 2 cancers-11-01702-f002:**
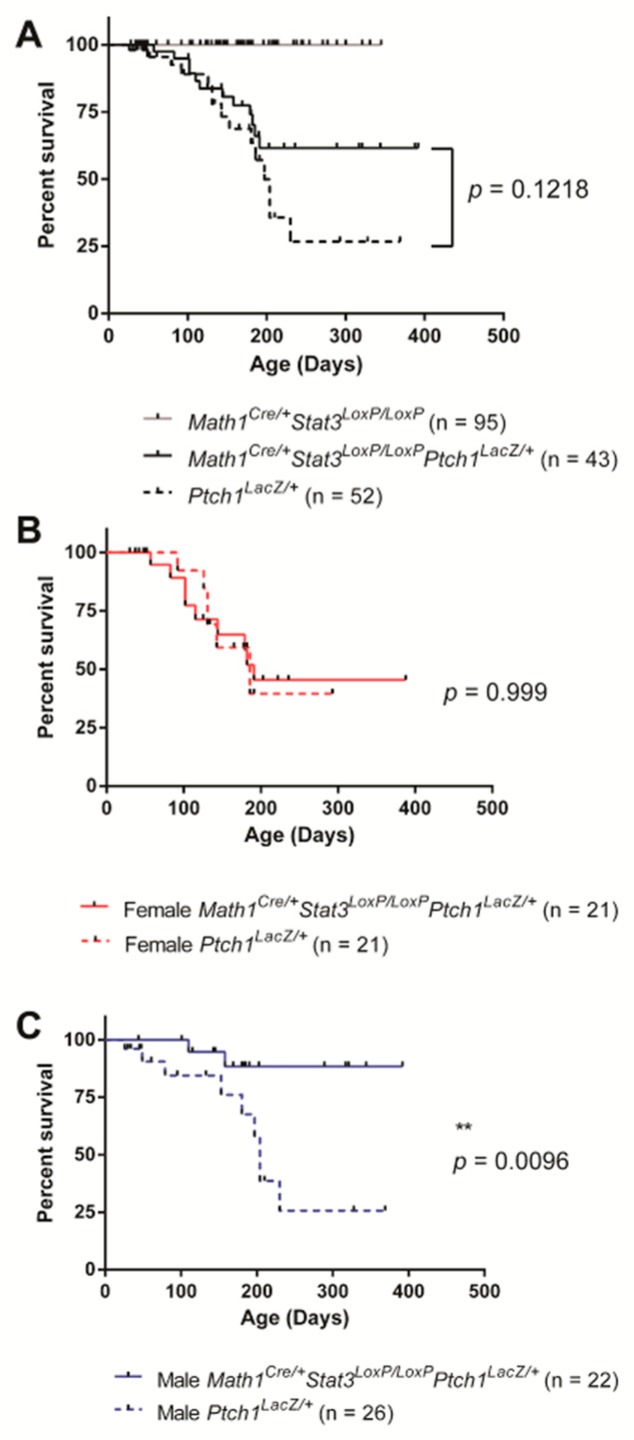
Stat3 deletion improved survival of male mice in the Ptch1^LacZ/+^ model of SHH MB. (**A**) Kaplan-Meier survival analysis of male and female mice. Ptch1^LacZ/+^ mice (n = 52) and Math1^Cre^Stat3^LoxP/LoxP^Ptch1^LacZ/+^ (n = 43) showed similar disease onset and overall survival (*p* = 0.1218). As expected, we observed no disease in mice lacking Stat3 from granule cell neurons if they were wild type for Ptch1 (Math1^Cre/+^Stat3^LoxP/LoxP^, n = 95). (**B**) Survival analysis of female Ptch1^LacZ/+^ mice (n = 21) and Math1^Cre^Stat3^LoxP/LoxP^Ptch1^LacZ/+^ (n = 21) showed similar survival (*p* = 0.999). (**C**) Survival of male Ptch1^LacZ/+^ mice (n = 26) and Math1^Cre^Stat3^LoxP/LoxP^Ptch1^LacZ/+^ (n = 22) mice showed that the loss of Stat3 provides significant protection from disease onset (*p* = 0.0096).

**Figure 3 cancers-11-01702-f003:**
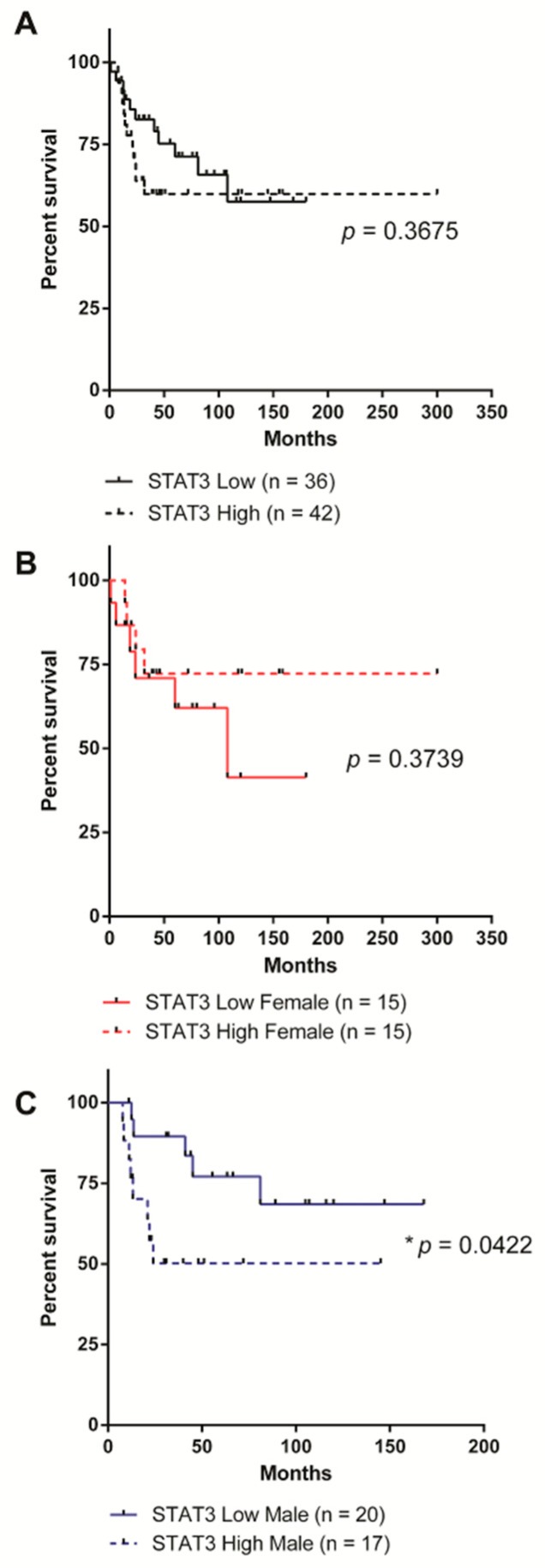
Low STAT3 expression correlates with improved survival in male SHH MB patients. (**A**) All patients were stratified by STAT3 levels prior to Kaplan-Meier survival analysis. Patients with low STAT3 (n = 36) and high STAT3 (n = 42) had similar survival (*p* = 0.3675). (**B**) When only female patients were analyzed, patients with high (n = 15) and low (n = 15) STAT3 still showed similar survival (*p* = 0.3739). (**C**) When only male patients were analyzed patients with low STAT3 (n = 20) showed significantly improved survival relative to those with high STAT3 (n = 17, *p* = 0.0422).

**Figure 4 cancers-11-01702-f004:**
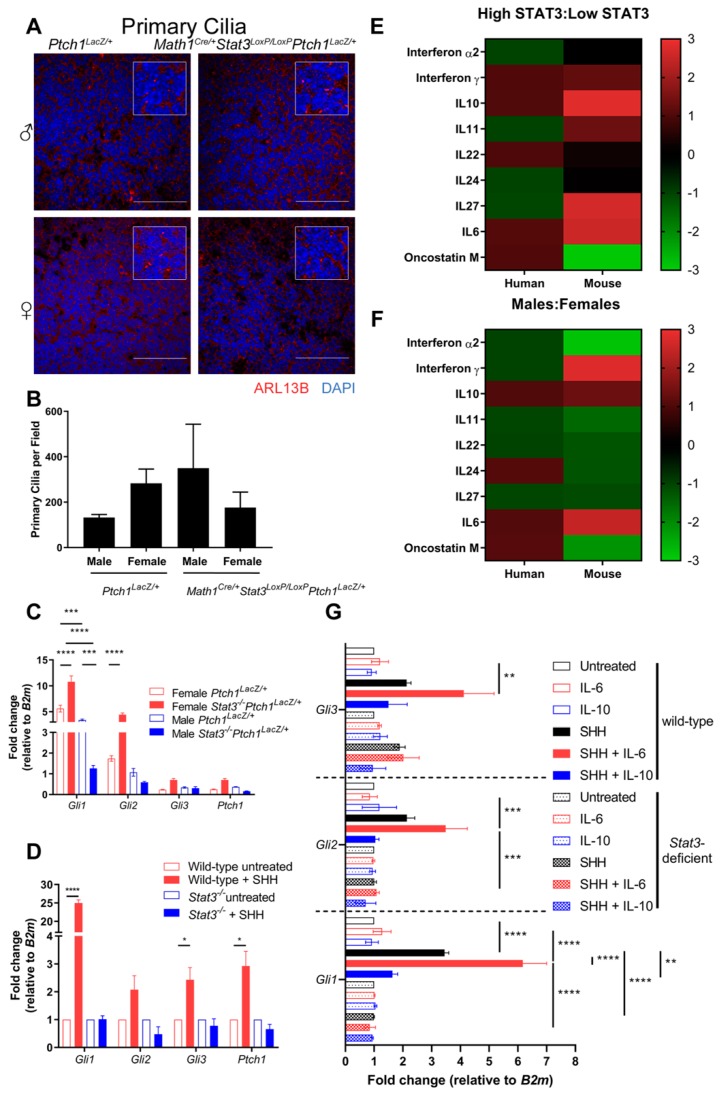
*Stat3* is required for SHH-mediated gene transcription. (**A**) Analysis of primary cilia formation by immunofluorescent staining cerebellum ARL13B and DAPI did not reveal significant differences in male or female *Ptch1^LacZ/+^*control or *Math1^Cre/+^Stat3^LoxP/LoxP^Ptch1^LacZ/+^* mice. (**B**) Cilia were quantified in three biological replicates (at least 100 cells per replicate) except for *Math1^Cre/+^Stat3^LoxP/LoxP^Ptch1^LacZ/+^* males where only duplicates were available). (**C)** qRT-PCR analysis for *Gli1*, *Gli2*, *Gli3* and *Ptch1* transcripts from tumor tissue taken from male and female mice of the indicated genotypes. (**D**) Wild-type and *Stat3^−/−^* MEFs were stimulated with SHH for 24 hours and expression of *Gli1*, *Gli2*, *Gli3* and *Ptch1* transcripts was measured by qRT-PCR. (**E**) High STAT3 vs. Low STAT3 heatmap. (**F**) Male vs. female heatmap. (**G**) qRT-PCR analysis for *Gli1*, *Gli2* and *Gli3* transcripts in WT and *Stat3*^−/−^ MEFs stimulated with IL-6, IL-10 and SHH Data are representative fields of immunohistochemistry or mean ± SEM of three biological replicates (* *p* < 0.05, ** *p* < 0.01, *** *p* < 0.001, **** *p* < 0.0001).

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
