# Peer review of "A Sexually Dimorphic Role for STAT3 in Sonic Hedgehog Medulloblastoma"

_cancers, 2019, doi:10.3390/cancers11111702_

Round 1

Reviewer 1 Report

In this study, the authors identify STAT3 as the molecular target driving sexual morphism in medulloblastoma using a mouse model of SHH MB with specific loss of STAT3 expression in the cell of origin.

Comments:

Figure 3. The conclusions drawn are not clear. In fig 3B, the curves for females survivors with high and low STAT3 do seem significantly different, contrary to the text description. Figure 4D. Since the authors speculate that STAT3 is regulating transcription of SHH target genes, they should check for pY levels or any typical STAT3 target gene levels upon treatment with SHH. Would it be possible to replicate some results in more classical medulloblastoma cell lines other than mefs?

Reviewer 2 Report

In my opinion this manuscript presents data, obtained with a good experimental scheme, useful in the pharmacological treatment of medulloblastoma.

I would describe the source of human tissues in more detail.

I would suggest adding the supplementary figures S1 S3 S4 to the manuscript.

Author Response

We thank the reviewer for the comments. The analysis of patient material was performed on RNA-Sequencing data published previously (Vanner, R.J et al Cancer Cell, 2014). We cited this reference and the accession numbers in the text of the manuscript and in the materials section. We have altered the text on page 6 to emphasize this point.

We agree with the reviewer that supplementary figures 1, 3 and 4 could be included in the main body of the manuscript. However, the space and figure limitations of brief communications necessitates their inclusion as supplemental data.

Reviewer 3 Report

In this manuscript the authors described a mechanism of higher incidence rate of medulloblastoma in male patients relative to females, particularly in SHH subgroup. The author specifically deleted Stat3 from granule cell precursors in a spontaneous model of mouse SHH medulloblastoma and found that it completely protects male, but not female mice from tumor initiation. They observed sex specific changes in IL-10 and IL-6 expression and show that IL-6 stimulation enhances SHH-mediated transcription of Gli in a STAT3-dependent manner and therefore claims that STAT3 may serve as a key molecule underpinning the sexual dimorphism in medulloblastoma. This is an interesting study with some potential significance. The results presented in the paper are potentially of interest to the readership of Cancers Journal. The research is reasonably designed and executed.

Comments:

What is the status of MYCN in Ptch1LacZ/+ Vs Math1Cre/+Stat3LoxP/LoxPPtch1LacZ/+ tumors as MYCN is known to be amplified in most of the SHH medulloblastoma.

In Fig 4D and 4G, what happens to Gli1 expression in the presence of SHH and pTyr705STAT3 inhibitor in WT MEF?

Page 12 line 306, sentence is incomplete.
